# Testing for Benford's Law in very small samples: Simulation study and a new test proposal

**Andrea Cerasa** *

European Commission, Joint Research Centre (JRC), Ispra, Italy

* andrea.cerasa@ec.europa.eu

## Abstract

Benford's Law defines a statistical distribution for the first and higher order digits in many datasets. Under very general condition, numbers are expected to naturally conform to the theorized digits pattern. On the other side, any deviation from the Benford distribution could identify an exogenous modification of the expected pattern, due to data manipulation or even fraud. Many statistical tests are available for assessing the Benford conformity of a sample. However, in some practical applications, the limited number of data to analyze may raise questions concerning their reliability. The first aim of this article is then to analyze and compare the behavior of Benford conformity testing procedures applied to very small samples through an extensive Monte Carlo experiment. Simulations will consider a thorough choice of compliance tests and a very heterogeneous selection of alternative distributions. Secondly, we will use the simulation results for defining a new testing procedure, based on the combination of three tests, that guarantees suitable levels of power in each alternative scenario. Finally, a practical application is provided, demonstrating how a sounding testing Benford compliance test for very small samples is important and profitable in anti-fraud investigations.

## Introduction

Benford's Law (BL) defines a probability distribution for patterns of significant digits in numerical data. Its formulation is grounded on the intriguing observation made first by Newcomb [1], then by Benford [2], who noticed a non-uniform amounts of wear in the pages of the logarithmic tables. In its complete form, the law states that the leading digits of many natural phenomena are not uniformly distributed, as one may expect, but follow a logarithmic distribution:

$$\mathbb{P}[D_1(X) = d_1, \ldots, D_m(X) = d_m] = \log_{10}\left(1 + \frac{1}{\sum_{l=1}^{m} 10^{m-l}d_l}\right) \qquad (1)$$

where $D_j(x)$ is the $j$-th significant digit of a positive real number $x$, $d_1 \in \{1, \ldots, 9\}$, and $d_j \in \{0, \ldots, 9\}$ for $j = 2, \ldots, m$. Focusing only on the first significant digit (FSD), expression (1) reduces to:

$$\mathbb{P}[D_1(X) = k] = p_k^B = \log_{10}(1 + 1/k), \quad \text{for } k = 1, 2, \ldots, 9. \qquad (2)$$

**Competing interests:** The authors have declared that no competing interests exist.

Theoretical studies investigated the properties and provided a limit theorem for the digit distribution. Classical references for this topic are [3–6]. At the same time, empirical applications have shown that many sets of numerical data are consistent with BL, at least in its simplest form Eq (2). Some examples in this sense are stock indexes [7], hydrology data [8] and volcanology data [9]. In addition, thanks to its generality and feasibility in different fields, the Benford distribution has fruitfully supported fraud investigations and the detection of manipulated data, in particular concerning COVID declared figures [10], scientific studies [11], media and social networks data [12] and international trade [13]. The assumption is that clean data (i.e. without any external manipulation) are distributed according to Eq (1). Generally, this condition is satisfied whenever numbers are the result of mathematical operations (multiplication, division, raising to power and so on) on values taken from different random variables, as in the case of accounting data [14]. The value of a purchase, for example, is the outcome of the multiplication between the number of items and their unitary price, which itself comes from the combination of different numbers. In such context, a significant departure from the theoretical distribution can point to data sets that include fabricated numbers.

The identification of non-Benford numbers can rely on a variety of statistical tests. Their empirical properties may significantly differ, but we expect that their power, ceteris paribus, increases with the sample dimension. On the contrary, when the number of observations is very small, "there may be insufficient power to meaningfully detect or confirm conformance with the law" ([15], page 2793). Nevertheless, in many practical applications, the number of figures available for each individual sample to test could be quite limited. In this case, the usual solution for increasing the expected sensitivity is to run the BL compliance tests on several individual samples merged together. Consider, for example, the investigations aimed to assess the digit distribution of the numbers published in scientific journals. Their analyses are usually not performed article by article, but they rather consider groups of articles pulled together (for example, those published in the same year, as in [16]). This strategy is not feasible however when the aim is to identify the specific individual sample that may contain irregularities. This is just the case of anti-fraud, where the target is to identify the economic operators that may have manipulated their declared numbers. Thus, in such contexts, the only strategy is to maximize the reliability of the expected outcome, that is to choose the testing procedure that, for a given significance level, guarantees the largest expected power against a wide range of possible alternatives.

In this article, we firstly analyze and compare the behavior of several testing procedures against a huge set of alternative scenarios. Besides considering a thorough choice of Benford compliance tests and a very heterogeneous selection of alternative distributions, the main novelty with respect to previous works with similar aim (as, for example [17–19]), is that we focus on samples with very small dimensions (i.e. $n = 20$). We do not expect to find a testing procedure that strictly dominates the other in all the different scenarios. The target is rather to find the ones that show the best general behavior independently of the alternative. Secondly, we use the testing performances obtained in the simulation exercise to derive and propose a combined test that guarantees suitable levels of power in each alternative scenario. The objective of this study is then twofold. From one side, we provide a comprehensive analysis of small samples properties of Benford's compliance tests that could support the researchers in the selection of the proper procedure. From the other, we propose an alternative method based on the combination of different tests that offers desirable performance against a wide range of possible deviations. An empirical application on international trade data shows how the availability of reliable BL testing procedures for very small samples can remarkably increase the range of applicability, and provide a valuable support for limiting the number of economic transactions that deserve further anti-fraud investigations.

The paper proceeds as follows. In the next section, the set of BL tests compared in the simulation exercise are briefly introduced, properly divided into three families: tests for the FSD, tests for the complete form of the BL and the summation test. This is followed by a section that describes the different alternative distributions of values considered in the Monte Carlo experiment. Therefore, the simulation results are presented and discussed. In addition, we used them to define a combined test with desirable empirical properties. We present an empirical application of Benford compliance tests on international trade data, stressing the importance of reliable methods for small samples in such context. The final section concludes.

## Testing the BL conformity

Many testing procedures allow to assess the conformity of a set of values to the Benford's theory. In the first part of our study, we want to study their behavior in small samples through a huge simulation exercise. We took into account most of the testing procedures proposed in past investigations, in order to provide a complete comparison. They can be divided into three families: tests for the FSD, tests for the complete BL, and the summation tests recently proposed by [20].

### Tests for the FSD

Given an $n$-sample of values $\{x_1, \ldots, x_n\}$, we can assess the BL conformity of the FSD through the null hypothesis:

$$H_0 : p_k = \log_{10}(1 + 1/k), \quad \text{for } k = 1, 2, \ldots, 9 \tag{3}$$

where $p_k = n_k/n$, and $n_k = \sum_{i=1}^{n} I(D_1(x_i) = k)$.

The first test is the well known Pearson goodness-of-fit test:

$$\chi^2 = \sum_{k=1}^{9} \frac{(n_k - np_k^B)^2}{np_k^B} = n \sum_{k=1}^{9} \frac{(p_k - p_k^B)^2}{p_k^B}. \tag{4}$$

that is asymptotically distributed as a $\chi_8^2$. It is surely one of most used statistics in BL compliance experiments, even though its potential low power, especially in small samples, suggests some cautions [21]. In this simulation exercise, it could be considered as a benchmark for assessing the gain in power associated with the other testing procedures.

A second category of FSD tests derives from the Cramer-Von-Mises, Watson and Anderson-Darling statistics, which are mainly used for testing the goodness-of-fit for continuous distributions [22, 23]. Let $F_k = \sum_{j=1}^{k} p_j$ and $F_k^B = \sum_{j=1}^{k} p_j^B$ denote the cumulative distributions of respectively the empirical and the expected proportions, whose difference is defined as $Z_k = F_k - F_k^B$. Define also the weights $w_k = (p_k^B + p_{k+1}^B)/2$ for $k = 1, \ldots, 8$ and $w_9 = (p_9^B + p_1^B)/2$, and use them for calculating the weighted mean of the cumulative distributions distances: $\bar{Z} = \sum_{k=1}^{9} w_k Z_k$. The three statistics are defined as:

$$W^2 = n \sum_{k=1}^{9} Z_k^2 w_k$$

$$U^2 = n \sum_{k=1}^{9} (Z_k - \bar{Z})^2 w_k \tag{5}$$

$$A^2 = n \sum_{k=1}^{8} \frac{Z_k^2 w_k}{F_k^B(1 - F_k^B)}$$

Asymptotic critical values are available for the three tests [19].

Then, we will consider two tests based on the Kolmogorov-Smirnov (KS) distance of the cumulative FSD distributions. In particular, the first is simply the KS deviation:

$$KS_d = \sqrt{n} \sup_{1 \le k \le 9} |F_k^B - F_k|. \tag{6}$$

whereas the second is the Kuiper test [24], defined as:

$$KU_d = \sup_{1 \le k \le 9} (F_k^B - F_k) + \sup_{1 \le k \le 9} (F_k - F_k^B). \tag{7}$$

Finally, the last FSD test considered is the Mean Test introduced by [16] and based on the simple observation that, if the FSD is distributed according to Eq (2), then its expected value is equal to 3.440 and its variance to 6.057. The Mean Test is defined as:

$$MT = \frac{|\sum_{k=1}^{9} p_k d_k - 3.440|}{\sqrt{6.057}}. \tag{8}$$

## Tests for the complete form of the BL

The second family of tests applied in the simulation experiment does not limit the attention to the FSD, but considers the whole digit distribution (1). Defining the significand of a value $y \in \mathbb{R}$ as $S(y) = 10^{\log_{10}|y| - \lfloor \log_{10}|y| \rfloor}$, with $\lfloor y \rfloor = \max\{m \in \mathbb{Z} : m \le y\}$ representing the floor function [5, 25], proved that:

$$Y \sim \text{Benford} \iff S(Y) \sim 10^U$$

where $U$ is a Uniform random variable on $[0; 1[$. Based on this theoretical result and ordering the observations $\{x_1, \ldots, x_n\}$ according to the value of the significand $\{S(x_{(1)}) \le \cdots \le S(x_{(n)})\}$, the BL conformity could be tested through the statistics based on the KS distance applied on the logarithm of the ordered significands. As before, we considered both the KS test:

$$KS_s = \max_{i \in [1,n]} |\log_{10} S(x_{(i)}) - i/n| \tag{9}$$

and the Kuiper test:

$$KU_s = \max_{i \in [1,n]} [i/n - \log_{10} S(x_{(i)})] + \sup_{i \in [1,n]} [\log_{10} S(x_{(i)}) - (i-1)/n]. \tag{10}$$

Recently [26], showed that $KS_s$ test for uniformity provides very good results for small, medium size and even large records.

In addition, we also consider the Anderson-Darling test [27], defined as:

$$AD = \sum_{i=1}^{n} \frac{2i-1}{n} [\ln(\log_{10} S(x_{(i)})) + \ln(1 - \log_{10} S(x_{(n+1-i)}))]. \tag{11}$$

**Summation test.**   Finally, the last test considered is the summation test $Q$ [20]. Starting from the definition of the significand previously introduced, we define $\mathcal{Z}_k(x_i) = S(x_i) \times I(FSD(x_i) = k)$ and $\bar{\mathcal{Z}}_k = n^{-1} \sum_1^n \mathcal{Z}_k(x_i)$. According to these definitions and representing $C = \log_{10} e$, the summation test limited to the first-digit case is given by:

$$Q = n(\bar{\mathcal{Z}} - \mu)' \Sigma^{-1} (\bar{\mathcal{Z}} - \mu) \tag{12}$$

where $\bar{\mathcal{Z}} = (\bar{\mathcal{Z}}_1, \ldots, \bar{\mathcal{Z}}_9)'$, $\mu$ is the nine elements vector $\mu = (C, \ldots, C)'$, and $\Sigma$ is the $(9 \times 9)$ matrix with elements $\sigma_{kk} = C(k + 1/2 - C)$ and $\sigma_{kj} = -C^2$ whenever $k \ne j$.

**Table 1. Alternative distributions considered in the simulation experiment.**

| Family | Parameter Space |
|---|---|
| GB | $\theta \in [-1.5, 1.5]$ |
| R | $\beta \in [-14, 10]$ |
| H | $\rho \in (0, 10]$ |
| LN | $X \sim Lognormal(\mu, \sigma)$ with $\mu \in \{0, 0.5, 1, 1.5, 2\}$, $\sigma \in [0.1, 1]$ |
| WB | $X \sim Weibull(a, b)$ with $a \in \{0.5, 1, 1.5, 2, 2.5\}$, $b \in [0.5, 5]$ |

## Description of the alternative distributions

The finite sample behavior of the eleven testing procedures introduced in the previous section are investigated in a simulation study. Table 1 presents all the alternatives distributions considered in the simulation experiment, together with the corresponding parameter space. In each simulation $s$, a sample $X^s = \{x_1^s, \ldots x_n^s\}$ is generated according to one of the pattern listed in the Table 1. Actually, the first three distributions are alternative patterns only for the FSD of $x$. In this case, the remaining digits of each number are simulated according to the Benford probabilities, in order to allow a suitable calculation of the tests for the complete form of the BL and of the summation test.

The first alternative family for the FSD is the Generalized Benford's Law (GB, [28, 29]):

$$\mathbb{P}[D_1(X) = k] = p_k^{GB}(\theta) = \begin{cases} \log_{10}\left(1 + \dfrac{1}{k}\right) & \theta = 0 \\ \dfrac{(k+1)^\theta - k^\theta}{10^\theta - 1} & \theta \neq 0 \end{cases} \tag{13}$$

where $k \in \{1, 2, \ldots, 9\}$ and $\theta \in \mathbb{R}$. Therefore, when $\theta = 0$, the GB reduces to the standard Benford distribution for the FSD (2), whereas for $\theta = 1$, the resulting distribution is uniform. This last case is quite interesting, since it mimics a manipulation strategy widely used in past applications [13, 19], based on a naive assignation of the first digit with equal probabilities. Finally, when $\theta < 0$, more weight is assigned to smaller digits (1 and 2), whereas when $\theta > 1$, more weight is assigned to bigger digits (from 4 to 9, see panel a of Fig 1).

Then, we considered the FSD patterns proposed by Rodriguez (R, [30]):

$$\mathbb{P}[D_1(X) = k] = p_k^R(\beta) = \begin{cases} \dfrac{1}{9}\left[1 + \dfrac{10}{9}\ln(10) + k\ln(k) - (k+1)\ln(k+1)\right] & \beta = 0 \\ \log_{10}\left(1 + \dfrac{1}{k}\right) & \beta = -1 \\ \dfrac{\beta+1}{9\beta} - \dfrac{(k+1)^{\beta+1} - k^{\beta+1}}{\beta(10^{\beta+1} - 1)} & \text{otherwise} \end{cases} \tag{14}$$

where $k \in \{1, 2, \ldots, 9\}$ and $\beta \in \mathbb{R}$. Again, the standard Benford distribution (2) and the uniform distribution of the FSD are two particular cases of this family. The former corresponds to $\beta = -1$, whereas the latter to $\beta = \pm\infty$. In general, when $\beta < 0$ ($\beta > 0$), the digit probabilities are decreasing according to a convex (concave) pattern (see panel b of Fig 1).

The last FSD distributions considered are the ones proposed by Hürlimann (H, [31]):

$$\mathbb{P}[D_1(X) = k] = p_k^H(\rho) = \frac{1}{2}\left\{[\log_{10}(1 + k)]^\rho - [\log_{10}k]^\rho - [1 - \log_{10}(1 + k)]^\rho + [1 - \log_{10}k]^\rho\right\} \tag{15}$$

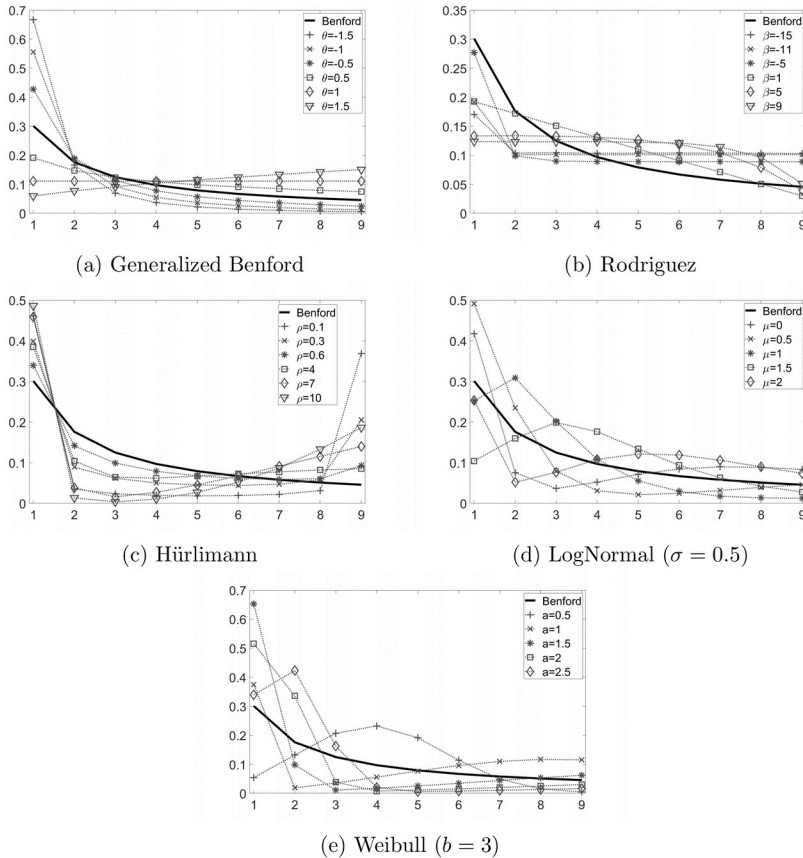

**Fig 1. FSD associated to the alternative distributions.**

with $k \in \{1, 2, \ldots, 9\}$ and $\rho > 0$. When $\rho = 1$ or $\rho = 2$, the distribution of the FSD corresponds to (2), whereas for other values of the parameter, the resulting distribution has a convex U-shape (see panel c of Fig 1).

The last two alternative distributions are the Lognormal (LN) and the Weibull (WB), two continuous random variables that may reasonably arise in many fields. Both of them can become very close to a Benford random variable for some combination of their parameters. Actually [4, 32], showed that a LN random variable with a large shape parameter is practically indistinguishable from a Benford random variable. This is why, in the simulation exercise, we decided to fix the scale parameter $\mu$ to five different values, and to let the shape parameter $\sigma$ vary from 0.1 to 1. Similar arguments apply also to the WB random samples. In this case, the simulated values closely fit the Benford distribution for small values of the shape parameter [6]. Again, we fixed scale parameter $a$ to five different values, and to let the shape parameter $b$ vary from 0.5 to 5. Panels d and e of Fig 1 provide a representation of the FSD probabilities associated to the LN and WB distributions when the shape parameter is respectively 0.5 and 3.

Therefore, the wide range of possible alternative distributions selected for the simulation exercise produces a heterogeneous set of patterns for the FSD probabilities. This allows an exhaustive assessment of the performance of the tests described in the previous section under very general contexts and scenarios.

**Table 2. Rejection rates (×100) when the Benford null (3) is true.**

| Test | GB(0) | R(-1) | H(1) | H(2) |
|---|---|---|---|---|
| $\chi^2$ | 1.027 | 0.986 | 1.034 | 1.034 |
| $W^2$ | 0.979 | 1.001 | 1.026 | 0.969 |
| $U^2$ | 0.999 | 1.041 | 1.003 | 0.990 |
| $A^2$ | 0.996 | 0.980 | 0.998 | 0.982 |
| $KS_d$ | 0.995 | 0.999 | 1.027 | 0.989 |
| $KU_d$ | 0.959 | 1.010 | 1.011 | 0.980 |
| $MT$ | 0.922 | 0.901 | 0.958 | 0.907 |
| $KS_s$ | 0.972 | 1.064 | 1.031 | 0.992 |
| $KU_s$ | 1.040 | 1.061 | 1.034 | 1.008 |
| $AD$ | 0.972 | 0.973 | 0.996 | 0.970 |
| $Q$ | 1.001 | 0.984 | 1.042 | 1.045 |

## Simulation results

Since the focus of our study is on small samples, for each alternative scenario we consider 4 sample dimensions, i.e. $n$ = 20, 30, 40 and 50. In this section, we present only the results obtained with $n$ = 20. The simulation outcomes for the other sample dimensions (available upon request) are numerically different, but confirm the patterns described for the smallest dimension. Even though the asymptotic critical values for some of the implemented tests are available, the small dimensions of the simulated samples recommends to use exact critical values. We then computed the 1% critical value for each test and each different sample size through one million simulations of Benford distributed values. For each alternative scenario listed in Table 1, we generate 100,000 simulations and calculate the rejection rates of each test.

The three alternative FSD patterns considered reduces to Eq (2) for particular values of its parameter. In these cases, rejection rates measure the empirical size of the tests. Table 2 presents the percentage rejection rates obtained when the FSD patterns corresponds exactly to Eq (2). The values range from 0.901% to 1.064% and confirm a suitable accuracy of the empirical size, not significantly different from the nominal value of 1%.

### Power against alternative FSD patterns

Table 3 shows the rejection rates of the tests against the three alternative FSD for a subset of the parameters considered in the Monte Carlo exercise (as before, the full set of results is available upon request). The panel reserved to GB alternatives highlights that, independently of the value of $\theta$, $A^2$, $W^2$, $AD$, $KS_d$ and $MT$ seem to offer the best general performances, whereas the power of $KS_s$ is optimal when $\theta < 0$, but deteriorates when $\theta > 0$. All the other tests seem to offer rejection rates always below average. When the digits are uniformly distributed (i.e. $\theta$ = 1), the power of $MT$, $W^2$ and $U^2$ is slightly less than 60%, and the one of $AD$ and $KS_d$ is close to 50%. All the other tests considered exhibit a power between 30 and 35%.

The central panel of Table 3 presents the rejection rates obtained with FSD simulated according to Eq (14). $A^2$, $W^2$, $AD$, $KS_d$ and $MT$ show again the best general properties independently of the value of the parameter, while the power of the remaining tests results smaller. The only exception is the case $\beta$ = 1, where the top performers are $U^2$ and $KU_d$.

Finally, the output against Hürlimann distributed digits is in the bottom panel of Table 3. In this case, the top performers are $U^2$, $KU_d$ and $KU_s$, whereas the power of $\chi^2$ and $Q$, is optimal only when $\rho < 1$. Instead, the rejection rates of $MT$, $KS_s$ and $W^2$ are always below average.

**Table 3. Rejection rates when the FSD is distribuited according to Eqs (13), (14) and (15).**

| Generalized Benford | | | | | | |
|---|---|---|---|---|---|---|
| $\theta$ | -1.5 | -1 | -0.5 | 0.5 | 1 | 1.5 |
| $\chi^2$ | 0.224 | 0.039 | 0.004 | 0.079 | 0.336 | 0.713 |
| $W^2$ | 0.907 | 0.543 | 0.115 | 0.127 | 0.573 | 0.924 |
| $U^2$ | 0.776 | 0.366 | 0.066 | 0.055 | 0.300 | 0.707 |
| $A^2$ | 0.896 | 0.519 | 0.103 | 0.137 | 0.593 | 0.931 |
| $KS_d$ | 0.848 | 0.463 | 0.094 | 0.110 | 0.498 | 0.874 |
| $KU_d$ | 0.690 | 0.287 | 0.048 | 0.072 | 0.380 | 0.788 |
| $MT$ | 0.826 | 0.413 | 0.070 | 0.145 | 0.592 | 0.926 |
| $KS_s$ | 0.888 | 0.538 | 0.131 | 0.052 | 0.339 | 0.761 |
| $KU_s$ | 0.689 | 0.304 | 0.060 | 0.050 | 0.283 | 0.677 |
| $AD$ | 0.832 | 0.430 | 0.080 | 0.112 | 0.534 | 0.906 |
| $Q$ | 0.280 | 0.069 | 0.010 | 0.069 | 0.301 | 0.670 |
| Rodriguez | | | | | | |
| $\beta$ | -15 | -11 | -5 | 1 | 5 | 9 |
| $\chi^2$ | 0.227 | 0.197 | 0.100 | 0.041 | 0.155 | 0.215 |
| $W^2$ | 0.353 | 0.286 | 0.096 | 0.044 | 0.281 | 0.394 |
| $U^2$ | 0.171 | 0.140 | 0.066 | 0.075 | 0.219 | 0.256 |
| $A^2$ | 0.384 | 0.321 | 0.124 | 0.037 | 0.259 | 0.378 |
| $KS_d$ | 0.325 | 0.272 | 0.110 | 0.045 | 0.249 | 0.343 |
| $KU_d$ | 0.230 | 0.188 | 0.077 | 0.068 | 0.227 | 0.287 |
| $MT$ | 0.413 | 0.356 | 0.152 | 0.021 | 0.189 | 0.322 |
| $KS_s$ | 0.196 | 0.156 | 0.053 | 0.018 | 0.140 | 0.212 |
| $KU_s$ | 0.169 | 0.139 | 0.065 | 0.054 | 0.179 | 0.224 |
| $AD$ | 0.346 | 0.290 | 0.115 | 0.021 | 0.187 | 0.299 |
| $Q$ | 0.203 | 0.178 | 0.092 | 0.035 | 0.133 | 0.188 |
| Hürlimann | | | | | | |
| $\rho$ | 0.1 | 0.3 | 0.6 | 4 | 7 | 10 |
| $\chi^2$ | 0.966 | 0.438 | 0.045 | 0.069 | 0.423 | 0.779 |
| $W^2$ | 0.387 | 0.079 | 0.020 | 0.038 | 0.140 | 0.274 |
| $U^2$ | 0.977 | 0.392 | 0.035 | 0.125 | 0.781 | 0.987 |
| $A^2$ | 0.863 | 0.232 | 0.031 | 0.054 | 0.273 | 0.587 |
| $KS_d$ | 0.751 | 0.192 | 0.024 | 0.058 | 0.285 | 0.508 |
| $KU_d$ | 0.978 | 0.379 | 0.034 | 0.104 | 0.692 | 0.965 |
| $MT$ | 0.390 | 0.149 | 0.034 | 0.052 | 0.166 | 0.266 |
| $KS_s$ | 0.455 | 0.085 | 0.018 | 0.043 | 0.187 | 0.341 |
| $KU_s$ | 0.940 | 0.283 | 0.026 | 0.088 | 0.622 | 0.936 |
| $AD$ | 0.844 | 0.254 | 0.032 | 0.057 | 0.294 | 0.579 |
| $Q$ | 0.967 | 0.437 | 0.045 | 0.073 | 0.429 | 0.774 |

## Power against continuous random variables

Table 4 presents the results obtained with data simulated according to a lognormal distribution with the five selected scale parameters. As expected, the rejection rates are inversely related to the shape parameter. When $\sigma$ decreases the rejection rates are close to 1, while when the shape approaches to 1, the rejection rates approach to the nominal level of the test. Even though every panel seems to tell a slightly different story, there are some testing procedures that seem to regularly offer desirable properties. In particular, they are $KU_s$, $U^2$ and $KU_d$. Also $KS_s$ and

**Table 4. Rejection rates for Lognormal distributed values.**

| $\mu$ | 0 | 0 | 0 | 0.5 | 0.5 | 0.5 |
|---|---|---|---|---|---|---|
| $\sigma$ | 0.3 | 0.5 | 0.7 | 0.3 | 0.5 | 0.7 |
| $\chi^2$ | 0.669 | 0.112 | 0.024 | 0.468 | 0.021 | 0.007 |
| $W^2$ | 0.219 | 0.057 | 0.019 | 0.985 | 0.337 | 0.044 |
| $U^2$ | 0.984 | 0.271 | 0.031 | 0.991 | 0.313 | 0.037 |
| $A^2$ | 0.428 | 0.078 | 0.023 | 0.976 | 0.296 | 0.037 |
| $KS_d$ | 0.439 | 0.101 | 0.023 | 0.991 | 0.356 | 0.044 |
| $KU_d$ | 0.957 | 0.216 | 0.028 | 0.986 | 0.278 | 0.031 |
| $MT$ | 0.207 | 0.067 | 0.023 | 0.855 | 0.172 | 0.022 |
| $KS_s$ | 0.502 | 0.107 | 0.025 | 0.996 | 0.434 | 0.064 |
| $KU_s$ | 0.993 | 0.309 | 0.035 | 0.992 | 0.312 | 0.036 |
| $AD$ | 0.858 | 0.150 | 0.030 | 0.964 | 0.256 | 0.034 |
| $Q$ | 0.801 | 0.125 | 0.024 | 0.547 | 0.039 | 0.009 |
| $\mu$ | 1 | 1 | 1 | 1.5 | 1.5 | 1.5 |
| $\sigma$ | 0.3 | 0.5 | 0.7 | 0.3 | 0.5 | 0.7 |
| $\chi^2$ | 0.623 | 0.048 | 0.007 | 0.859 | 0.142 | 0.026 |
| $W^2$ | 0.355 | 0.025 | 0.009 | 0.918 | 0.176 | 0.023 |
| $U^2$ | 0.940 | 0.212 | 0.027 | 0.998 | 0.397 | 0.045 |
| $A^2$ | 0.342 | 0.020 | 0.008 | 0.889 | 0.141 | 0.020 |
| $KS_d$ | 0.415 | 0.049 | 0.012 | 0.889 | 0.195 | 0.025 |
| $KU_d$ | 0.910 | 0.201 | 0.025 | 0.987 | 0.314 | 0.041 |
| $MT$ | 0.054 | 0.025 | 0.008 | 0.015 | 0.033 | 0.014 |
| $KS_s$ | 0.654 | 0.089 | 0.018 | 0.855 | 0.109 | 0.010 |
| $KU_s$ | 0.993 | 0.314 | 0.035 | 0.992 | 0.309 | 0.035 |
| $AD$ | 0.064 | 0.005 | 0.005 | 0.658 | 0.074 | 0.013 |
| $Q$ | 0.645 | 0.052 | 0.008 | 0.836 | 0.129 | 0.023 |
| $\mu$ | 2 | 2 | 2 | | | |
| $\sigma$ | 0.3 | 0.5 | 0.7 | | | |
| $\chi^2$ | 0.852 | 0.180 | 0.035 | | | |
| $W^2$ | 0.904 | 0.225 | 0.035 | | | |
| $U^2$ | 0.936 | 0.182 | 0.023 | | | |
| $A^2$ | 0.909 | 0.235 | 0.039 | | | |
| $KS_d$ | 0.944 | 0.260 | 0.039 | | | |
| $KU_d$ | 0.921 | 0.215 | 0.031 | | | |
| $MT$ | 0.909 | 0.256 | 0.045 | | | |
| $KS_s$ | 0.901 | 0.172 | 0.021 | | | |
| $KU_s$ | 0.992 | 0.308 | 0.036 | | | |
| $AD$ | 0.942 | 0.231 | 0.036 | | | |
| $Q$ | 0.870 | 0.176 | 0.033 | | | |

$KS_d$ yield adequate results on average. On the other side of the ranking, what stands out are the poor performances offered by $W^2$ when $\mu = 0$; by $MT$ when $\mu = 0$, 1 and 1.5; by $AD$ when $\mu = 1$; and by $\chi^2$ and $Q$ when $\mu = 0.5$.

Table 5 shows instead the rejection rates obtained with Weibull distributed values for fixed scale coefficients. As expected, now the rejection rates are directly related to the shape parameter. Thus, when $b$ increases, the power approaches to 1. Again, the outstanding performers are $KU_s$, $U^2$ and $KU_d$, followed by $KS_s$ and $KS_d$ that also offer outcomes above the average. Particularly negative results are showed by $MT$ when $a = 0.5$ and 1.5; and by $\chi^2$ and $Q$ when $a = 2$.

**Table 5. Rejection rates for Weibull distributed values.**

| $a$ | 0.5 | 0.5 | 0.5 | 1 | 1 | 1 |
|---|---|---|---|---|---|---|
| $b$ | 2 | 3 | 4 | 2 | 3 | 4 |
| $\chi^2$ | 0.111 | 0.507 | 0.891 | 0.085 | 0.329 | 0.615 |
| $W^2$ | 0.149 | 0.488 | 0.834 | 0.050 | 0.183 | 0.356 |
| $U^2$ | 0.279 | 0.881 | 0.995 | 0.141 | 0.533 | 0.864 |
| $A^2$ | 0.121 | 0.429 | 0.804 | 0.068 | 0.267 | 0.514 |
| $KS_d$ | 0.154 | 0.484 | 0.798 | 0.078 | 0.294 | 0.533 |
| $KU_d$ | 0.233 | 0.821 | 0.991 | 0.118 | 0.425 | 0.749 |
| $MT$ | 0.038 | 0.012 | 0.002 | 0.075 | 0.274 | 0.471 |
| $KS_s$ | 0.082 | 0.345 | 0.687 | 0.072 | 0.269 | 0.531 |
| $KU_s$ | 0.214 | 0.821 | 0.992 | 0.213 | 0.822 | 0.993 |
| $AD$ | 0.068 | 0.230 | 0.577 | 0.115 | 0.534 | 0.912 |
| $Q$ | 0.103 | 0.481 | 0.876 | 0.093 | 0.441 | 0.826 |
| $a$ | 1.5 | 1.5 | 1.5 | 2 | 2 | 2 |
| $b$ | 2 | 3 | 4 | 2 | 3 | 4 |
| $\chi^2$ | 0.028 | 0.296 | 0.789 | 0.014 | 0.113 | 0.372 |
| $W^2$ | 0.176 | 0.634 | 0.936 | 0.191 | 0.703 | 0.948 |
| $U^2$ | 0.263 | 0.870 | 0.997 | 0.153 | 0.714 | 0.969 |
| $A^2$ | 0.155 | 0.610 | 0.934 | 0.161 | 0.640 | 0.921 |
| $KS_d$ | 0.183 | 0.649 | 0.949 | 0.239 | 0.841 | 0.992 |
| $KU_d$ | 0.205 | 0.788 | 0.989 | 0.147 | 0.738 | 0.981 |
| $MT$ | 0.052 | 0.179 | 0.412 | 0.115 | 0.469 | 0.783 |
| $KS_s$ | 0.261 | 0.780 | 0.976 | 0.295 | 0.872 | 0.995 |
| $KU_s$ | 0.213 | 0.821 | 0.993 | 0.212 | 0.821 | 0.992 |
| $AD$ | 0.156 | 0.604 | 0.920 | 0.117 | 0.548 | 0.887 |
| $Q$ | 0.040 | 0.332 | 0.803 | 0.024 | 0.182 | 0.477 |
| $a$ | 2.5 | 2.5 | 2.5 | | | |
| $b$ | 2 | 3 | 4 | | | |
| $\chi^2$ | 0.024 | 0.214 | 0.596 | | | |
| $W^2$ | 0.084 | 0.492 | 0.869 | | | |
| $U^2$ | 0.117 | 0.606 | 0.937 | | | |
| $A^2$ | 0.068 | 0.412 | 0.801 | | | |
| $KS_d$ | 0.122 | 0.675 | 0.957 | | | |
| $KU_d$ | 0.106 | 0.557 | 0.915 | | | |
| $MT$ | 0.075 | 0.377 | 0.663 | | | |
| $KS_s$ | 0.179 | 0.785 | 0.986 | | | |
| $KU_s$ | 0.211 | 0.821 | 0.992 | | | |
| $AD$ | 0.035 | 0.236 | 0.618 | | | |
| $Q$ | 0.030 | 0.244 | 0.649 | | | |

## General comments

The comparison of different BL compliance tests in a simulated environment provided several important conclusions. First of all, it is not necessarily true that in very small samples there is insufficient power to meaningfully detect or confirm conformance with the BL. Simulations showed that a careful choice of the testing procedure allows to significantly increase the expected power. Consider, for example, the case of the uniform alternative. As previously stressed, this is the FSD distribution that we should expect when numbers are falsified by a

manipulator that believes that the proportion of ones, twos,. . ., nines in the first digit should be equal. By choosing the usual $\chi^2$ test, as most of the analysts do in practice, we can expect a power of 33.5%. But if we consider $A^2$ or $MT$ the expected power almost doubles.

Also in other scenarios, simulation results confirmed that the power of $\chi^2$ is usually dominated by the one offered by alternative testing procedures. Thus, despite its popularity, the use of this test should be avoided, at least in very small samples. The other testing procedures considered in the simulation exercise showed very different characteristics, and there is not one that strictly dominates the others. It is also interesting to note that the statistics aimed to test the simplest null (2) work properly also when the deviation affects all the digits, and are not necessary dominated by their counterparts that consider the complete joint digits distribution. This is, for example, the case of $U^2$, which is optimal not only when the FSD follows a Hürlimann distribution, but also when the simulated values are Weibull distributed with scale parameter equal to 0.5. However, it performs poorly when the FSD is simulated according to a Generalized Benford distribution with $\theta > 0$ and to a Rodriguez distribution with $\beta < -1$. This sub-optimal behavior is common also to other testing procedures. Therefore, it is not possible to identify a test that is able to offer suitable expected performances independently of the alternative scenario. One alternative option could be to define a combined test that merges together the positive features offered by the single tests, as described in the following section.

## Combining tests for improving the general performance

The simulation results provided an exhaustive of the small sample behavior of different BL conformity tests. In particular, two of them offered the best general results across the different alternative scenarios: $KU_s$ and $U^2$. The idea is then to combine the $p$-values of these tests in order to derive a procedure that offers a desirable level of power in all the alternative scenarios proposed. However, both offered poor performances when the FSD was distributed according to the Generalized Benford and the Rodriguez distribution. Therefore, in the definition of the combined test, we decided to consider also the $A^2$ test, whose performances under the Generalized Benford and the Rodriguez alternatives are optimal. Among the possible choices for combining the $p$-values, we chose the minimum function (see [33] for more details on this issue). Thus, the statistics is given by:

$$\Gamma_{\{KU_s, U^2, A^2\}} = 1 - \min\{\pi(KU_s), \pi(U^2), \pi(A^2)\} \tag{16}$$

where $\pi(\cdot)$ is the $p$-value function, defined as $1 - F(\cdot)$. As usual, the $p$-values of the single tests and the 1% critical values of the combined test $\Gamma$ will be calculated through the one million simulations of Benford distributed numbers. Our expectation is that the performances of $\Gamma$ represent a favorable compromise of the ones of $KU_s$, $U^2$ and $A^2$.

Figs 2–4 provide an immediate assessment of the behavior of $\Gamma$ with respect to the three single tests. In addition, they allow a direct comparison with the maximum and the minimum rejection rates obtained with all the tests considered in the simulation exercise. The first thing to notice is that the combined test is always the second best choice, independently of which of the three single tests was the best choice. Secondly, the power of $\Gamma$ is always closer to the best among $KU_s$, $U^2$ and $A^2$, than to the worst. These two features allow the combined test to offer convincing outcomes even when one or two single tests perform below the average, Consider, for example, the case represented in **PowerCombinedGB** for $\theta > 0$: the power of $KU_s$, $U^2$ is very close to the absolute minimum, whereas the rejection rate of $\Gamma$ is very close to the one of $A^2$, which is the absolute maximum. In conclusion, test $\Gamma$ provides a convenient trade-off of the behaviors of the three single tests in all the scenarios considered.

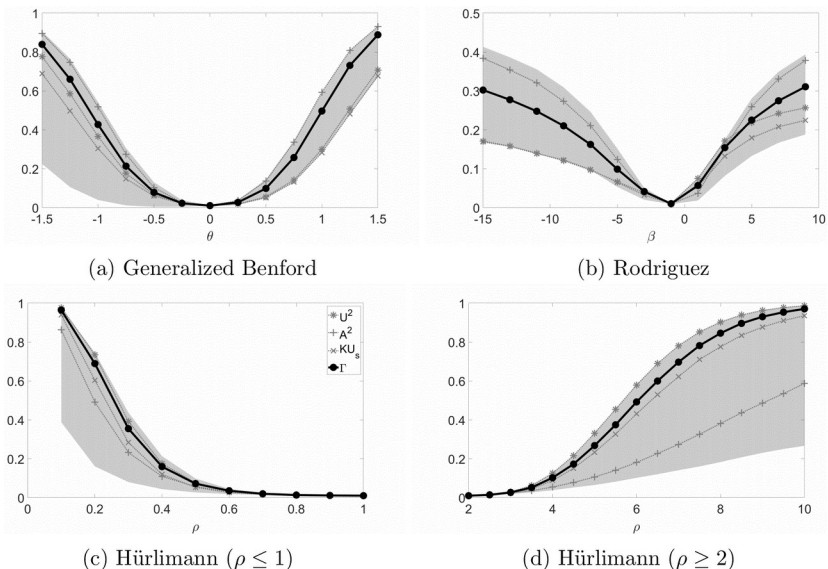

(a) Generalized Benford

(b) Rodriguez

(c) Hürlimann ($\rho \leq 1$)

(d) Hürlimann ($\rho \geq 2$)

**Fig 2. Rejection rates of the combined test for alternative FSD.** The gray region represents the area between the maximum and the minimum value of the power obtained in the simulations.

## Empirical application on international trade data

In this section, we provide some examples of the application of the BL compliance tests on Customs data. The contrast of fraud in this context is crucial. The protection of the financial interests is indeed a fundamental task of the European Union (EU) administration, and the

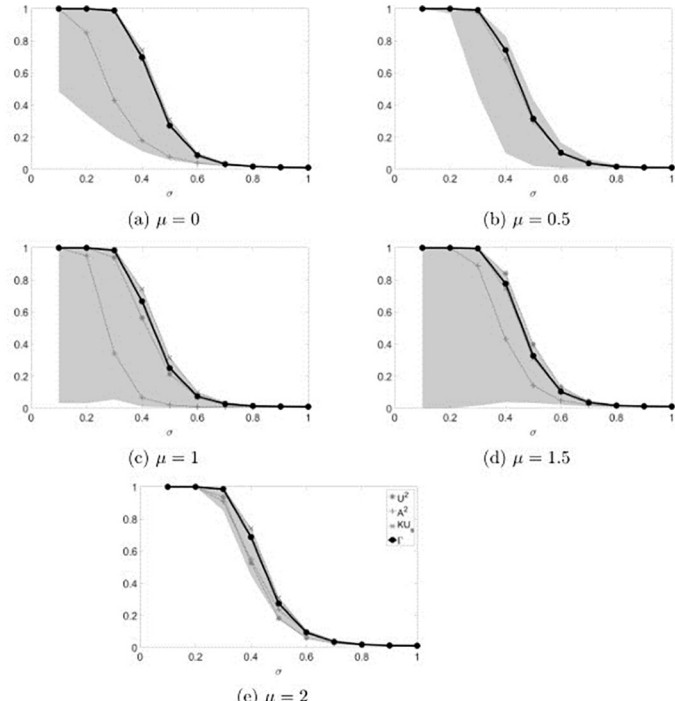

(a) $\mu = 0$

(b) $\mu = 0.5$

(c) $\mu = 1$

(d) $\mu = 1.5$

(e) $\mu = 2$

**Fig 3. Rejection rates of the combined test for lognormal distributed values.** The gray region represents the area between the maximum and the minimum value of the power obtained in the simulations.

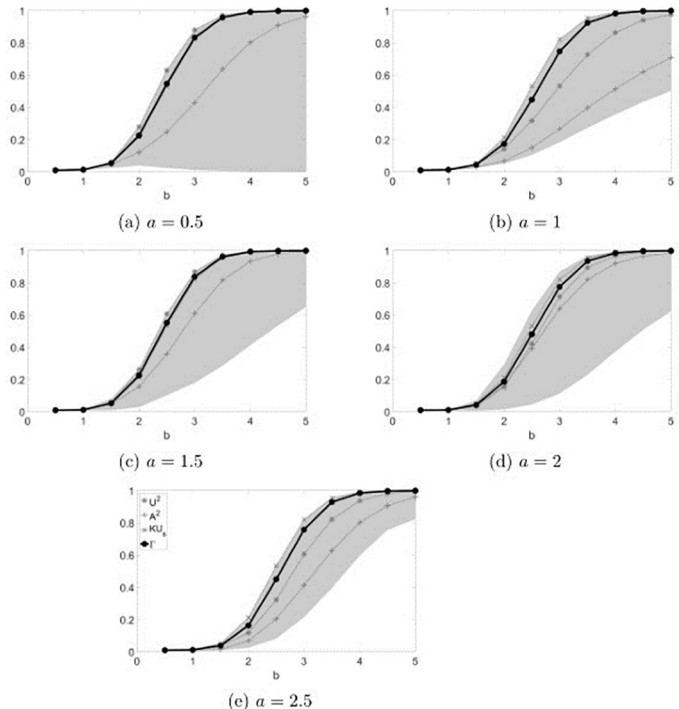

**Fig 4. Rejection rates of the combined test for Weibull distributed values.** The gray region represents the area between the maximum and the minimum value of the power obtained in the simulations.

collection of customs duties on imports represents the principal source of traditional own resources of the EU budget [34]. In 2017, the EU's revenue from customs duties was more than 20 million, representing almost 15% of EU total revenue (source: https://ec.europa.eu/budget/library/biblio/publications/2018/financial-report_en.pdf). Under-reporting the value of the imports is usually the main strategy pursued to pay less duties or excises, or to evade import restrictions and certain anti-dumping measures [35]. Proper statistical analyses provide support for the identification of anomalous trades that may result from unfair commercial strategies [36, 37]. Several studies have shown that statistical methods based also on the Benford Law can be profitably used in this field, aiming to spot traders with more than 50 transactions that are suspected of data manipulation [13, 20, 38]. The possibility to extend the focus also to economic operators with less imports can significantly increase the feasibility of the anti-fraud analysis, especially when the investigations are limited to particular periods, or to particular groups of products. Just to have an idea, Table 6 shows the distribution of traders according to the number of declared imports during the whole 2014 in a single Member State of the European Union (not revealed for confidentiality issues). Considering that one year is the usual period length of Customs audit, it is evident how extending the focus also to less active importers allows to almost double the number of economic operators we can monitor. In addition, a reliable and accurate testing procedure for small samples allows to tighten the number of suspicious imports to investigate, reducing the costs of the demanding and time consuming controls necessary to prove the eventual fraud.

Consider, for instance, the two examples of the application of the BL compliance tests represented in Table 7. Data were collected in the context of a specific operation by a Member State of the EU and, for confidentiality reasons, the two traders will be labeled simply as Trader 1 and Trader 2. Trader 1 collected 58 imports during three years. The p-values of the tests a

**Table 6. Number of imports per trader in 2014 for an EU member state.**

| Number of imports per year | Number of traders |
|---|---|
| less than 20 imports | 210,521 |
| between 20 and 29 imports | 6,627 |
| between 30 and 39 imports | 3,962 |
| between 40 and 49 imports | 2,714 |
| 50 or more imports | 14,459 |

suggest a potential departure of the values declared from the theoretical distribution. Most of the p-values are indeed very close or even smaller than 1%, with only one of them larger than 5%. A year-by-year analysis reveals that the 21 import values declared in 2015 yield the smallest p-value, and should therefore be the first to investigate in the search for frauds. Actually, one of those 21 imports was checked by Custom authority, and the corresponding import resulted to be under-valued. The analysis of Trader 2 provides another interesting pattern. Also in this case, the 94 imports were traded in a three years period, and the p-values of all the tests

**Table 7. P-values of Benford tests applied on the import values declared by two traders.**

| | Trader 1 | | | |
|---|---|---|---|---|
| Period | 2013–2015 | 2013 | 2014 | 2015 |
| $n$ | 58 | 18 | 19 | 21 |
| $\chi^2$ | 0.010 | 0.025 | 0.144 | 0.018 |
| $W^2$ | 0.020 | 0.065 | 0.697 | 0.034 |
| $U^2$ | 0.000 | 0.009 | 0.516 | 0.003 |
| $A^2$ | 0.020 | 0.079 | 0.558 | 0.037 |
| $KS_d$ | 0.031 | 0.088 | 0.406 | 0.050 |
| $KU_d$ | 0.001 | 0.006 | 0.279 | 0.007 |
| $MT$ | 0.619 | 0.481 | 0.546 | 0.454 |
| $KS_s$ | 0.019 | 0.092 | 0.281 | 0.061 |
| $KU_s$ | 0.000 | 0.006 | 0.420 | 0.002 |
| $AD$ | 0.041 | 0.135 | 0.732 | 0.119 |
| $Q$ | 0.014 | 0.023 | 0.176 | 0.023 |
| $\Gamma_{\{KU_s, U^2, A^2\}}$ | 0.000 | 0.013 | 0.585 | 0.004 |
| | Trader 2 | | | |
| Period | 2012–2014 | 2012 | 2013 | 2014 |
| $n$ | 94 | 31 | 26 | 37 |
| $\chi^2$ | 0.735 | 0.412 | 0.870 | 0.013 |
| $W^2$ | 0.155 | 0.125 | 0.774 | 0.000 |
| $U^2$ | 0.536 | 0.233 | 0.894 | 0.002 |
| $A^2$ | 0.145 | 0.147 | 0.784 | 0.000 |
| $KS_d$ | 0.322 | 0.241 | 0.745 | 0.001 |
| $KU_d$ | 0.574 | 0.350 | 0.935 | 0.002 |
| $MT$ | 0.111 | 0.177 | 0.455 | 0.002 |
| $KS_s$ | 0.225 | 0.273 | 0.484 | 0.001 |
| $KU_s$ | 0.617 | 0.398 | 0.780 | 0.011 |
| $AD$ | 0.231 | 0.164 | 0.878 | 0.001 |
| $Q$ | 0.758 | 0.457 | 0.925 | 0.018 |
| $\Gamma_{\{KU_s, U^2, A^2\}}$ | 0.247 | 0.248 | 0.898 | 0.001 |

suggest BL compliance of the declared values. However, the year-by-year analysis identifies a significant deviation from the Benford distribution for the 37 values declared in 2014. Actually, 6 of them were checked by Custom authority, and the corresponding imports resulted to be under-valuated.

In conclusion, the availability of suitable and powerful procedures for testing BL compliance in small samples can efficiently support the Custom anti-fraud activities. It allows not only to extend the range of applicability through the possibility of use them for traders with few operations, but also to identify small subsets of imports that deserve further investigations.

## Conclusion

The popularity of the BL is remarkably increasing in these last years. Since a set of numbers is expected to be Benford compliant under very general conditions, it can be used in many applications where the target is to identify potentially manipulated figures, as, for example, anti-fraud investigations. The automatic assessment of BL conformity through statistical methods requires testing procedures with desirable statistical properties. Especially in audit where many samples are tested, we aim for a statistical test which controls the number of false alarms and guarantees at the same time a suitable level of power. The choice of the Benford compliance test is even more important when the number of observations in the sample is small, given that suitable level of powers are very difficult to achieve.

The aim of this article was first of all to provide an extensive analysis of the performances of several BL compliance tests in very small samples. Simulation results proved that small sample properties can significantly vary, depending on the alternative scenario. In general, every test alternates good and average results across all the alternatives and there is no a procedure that strictly dominates the others. However, the following regularities emerged from the simulation exercise:

- despite its popularity, the power of Pearson's $\chi^2$ test was often below average;

- $KU_s$ and $U^2$ achieved outstanding results in most of the alternative scenarios considered, but not when the FSD is uniformly distributed.

The need of a procedure with optimal results independently of the alternative digits pattern encouraged us to define a new test. The main idea was to merge the positive small sample properties offered by some of the tests in the simulation experiment. Therefore we proposed to combines the $p$-values of $KU_s$, $U^2$ and $A^2$ through the *min* function. The resulting test achieved always desirable levels of powers in all alternative designs, even when one or two single tests performed below the average.

Finally, an empirical application presented a practical case where the availability of a reliable testing procedure for small samples allows *(i)* to increase the number of samples under investigation; and *(ii)* a more accurate selection of the cases that are suspected of data manipulation. This second issue is particularly relevant in anti-fraud audits. Non-conformity to the BL does not necessarily mean that the corresponding economic operator is a fraudster. Further, sometimes expensive, investigations are required to prove the irregularities. Therefore, having the possibility to focus the attention on a restricted number of suspect cases together with a reliable control over the number of false alarms are two essential requirements for an affordable anti-fraud analysis.

## Supporting information

**S1 File.**
(ZIP)

## Author Contributions

**Conceptualization:** Andrea Cerasa.

**Data curation:** Andrea Cerasa.

**Formal analysis:** Andrea Cerasa.

**Investigation:** Andrea Cerasa.

**Methodology:** Andrea Cerasa.

**Software:** Andrea Cerasa.

**Supervision:** Andrea Cerasa.

**Writing – original draft:** Andrea Cerasa.

**Writing – review & editing:** Andrea Cerasa.

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
