## [Decision Letter · Decision Letter 0]

1 Jun 2022

PONE-D-21-17178Testing for Benford's Law in very small samples: simulation study and a new test proposalPLOS ONE

Dear Dr. Cerasa,

Thank you for submitting your manuscript to PLOS ONE. After careful consideration, we feel that it has merit but does not fully meet PLOS ONE’s publication criteria as it currently stands. Therefore, we invite you to submit a revised version of the manuscript that addresses the points raised during the review process. I have received two reports for this paper. Both reviewers seem to be happy with the paper. They have made only minor suggestions for the paper. Please handle them carefully.

We look forward to receiving your revised manuscript.

Kind regards,

Miguel A. Fernández, Ph.D.

Academic Editor

PLOS ONE

Journal Requirements:

Reviewers' comments:

Reviewer's Responses to Questions

**Comments to the Author**

1. Is the manuscript technically sound, and do the data support the conclusions?

Reviewer #1: Yes

Reviewer #2: Yes

2. Has the statistical analysis been performed appropriately and rigorously? 

Reviewer #1: Yes

Reviewer #2: Yes

3. Have the authors made all data underlying the findings in their manuscript fully available?

Reviewer #1: Yes

Reviewer #2: Yes

4. Is the manuscript presented in an intelligible fashion and written in standard English?

Reviewer #1: Yes

Reviewer #2: Yes

5. Review Comments to the Author

Reviewer #1: Introducing new Benford tests that work for small and/or large datasets is welcome as the existing tests usually do not meet the expectations. In this regards, I recommend publication of this work. However, one aspect of any good research is exploring and citing relevant work. For example, one work I am aware of which has not been cited in the paper under review is the following:

Benford Test Based on Logarithmic Property

by Javad Kazemitabar, Jalil Kazemitabar

International Journal of Auditing Technology

DOI: 10.1504/IJAUDIT.2022.10047587

As a reviewer, I require this change to be made.

Reviewer #2: A few minor suggestions/corrections: p 10 line 198 and figures 1 and 2, umlaut ü on Hürlimann; p 12 line 221 "not necessarily"; p 16 line 282 "almost 15% of the EU"; references 3 and 4 have no "et al"; references 1,2,14,26,30, 30 are missing capitals on journal titles; reference 32 should read "combining p-values".

6. PLOS authors have the option to publish the peer review history of their article (what does this mean?). If published, this will include your full peer review and any attached files.

Reviewer #1: No

Reviewer #2: No

---

## [Author Response · Author response to Decision Letter 0]

30 Jun 2022

Reviewer #1: Introducing new Benford tests that work for small and/or large datasets is welcome as the existing tests usually do not meet the expectations. In this regards, I recommend publication of this work. However, one aspect of any good research is exploring and citing relevant work. For example, one work I am aware of which has not been cited in the paper under review is the following: 

Benford Test Based on Logarithmic Property by Javad Kazemitabar, Jalil Kazemitabar

International Journal of Auditing Technology, DOI: 10.1504/IJAUDIT.2022.10047587

As a reviewer, I require this change to be made.

Thanks a lot for your interesting bibliographic suggestion. It was added to the revised version of the manuscript.

Reviewer #2: A few minor suggestions/corrections: p 10 line 198 and figures 1 and 2, umlaut ü on Hürlimann; p 12 line 221 "not necessarily"; p 16 line 282 "almost 15% of the EU"; references 3 and 4 have no "et al"; references 1,2,14,26,30, 30 are missing capitals on journal titles; reference 32 should read "combining p-values".

Thanks a lot for spotting these mistakes. They are all fixed in the new version of the manuscript.

---

## [Decision Letter · Decision Letter 1]

12 Jul 2022

Testing for Benford's Law in very small samples: simulation study and a new test proposal

PONE-D-21-17178R1

Dear Dr. Cerasa,

We’re pleased to inform you that your manuscript has been judged scientifically suitable for publication and will be formally accepted for publication once it meets all outstanding technical requirements.

Kind regards,

Miguel A. Fernández, Ph.D.

Academic Editor

PLOS ONE

Additional Editor Comments (optional):

Reviewers' comments:

Reviewer's Responses to Questions

**Comments to the Author**

1. If the authors have adequately addressed your comments raised in a previous round of review and you feel that this manuscript is now acceptable for publication, you may indicate that here to bypass the “Comments to the Author” section, enter your conflict of interest statement in the “Confidential to Editor” section, and submit your "Accept" recommendation.

Reviewer #1: All comments have been addressed

Reviewer #2: All comments have been addressed

2. Is the manuscript technically sound, and do the data support the conclusions?

Reviewer #1: Yes

Reviewer #2: Yes

3. Has the statistical analysis been performed appropriately and rigorously? 

Reviewer #1: Yes

Reviewer #2: Yes

4. Have the authors made all data underlying the findings in their manuscript fully available?

Reviewer #1: Yes

Reviewer #2: Yes

5. Is the manuscript presented in an intelligible fashion and written in standard English?

Reviewer #1: Yes

Reviewer #2: Yes

6. Review Comments to the Author

Reviewer #1: All my concerns are addressed. The specific topic covered in this paper is important in the sense that many of the existing Benford tests are not general enough. In other words, they work in some scenarios such as medium size dataset, but fail to cover small or large datasets. As such I welcome the manuscript under review.

Reviewer #2: (No Response)

7. PLOS authors have the option to publish the peer review history of their article (what does this mean?). If published, this will include your full peer review and any attached files.

Reviewer #1: **Yes: **Javad Kazemitabar

Reviewer #2: No

---

## [Editor Report · Acceptance letter]

14 Jul 2022

PONE-D-21-17178R1 

Testing for Benford’s Law in very small samples: simulation study and a new test proposal 

Dear Dr. Cerasa:

I'm pleased to inform you that your manuscript has been deemed suitable for publication in PLOS ONE. Congratulations! Your manuscript is now with our production department. 

Kind regards, 

on behalf of

Dr Miguel A. Fernández 

Academic Editor

PLOS ONE